# The Molecular Mechanisms and Function of miR-15a/16 Dysregulation in Fibrotic Diseases

**DOI:** 10.3390/ijms232416041

**Published:** 2022-12-16

**Authors:** Dada Wen, Huamin Zhang, Yutong Zhou, Jie Wang

**Affiliations:** Department of Immunology, Xiangya School of Medicine, Central South University, Changsha 410078, China

**Keywords:** miR-15a, miR-16, biological characteristics, fibrosis, functional mechanism

## Abstract

MicroRNAs (miRNAs) are a class of short, endogenous, non-coding, single-stranded RNAs that can negatively regulate the post-transcriptional expression of target genes. Among them, miR-15a/16 is involved in the regulation of the occurrence and development of fibrosis in the liver, lungs, heart, kidneys, and other organs, as well as systemic fibrotic diseases, affecting important cellular functions, such as cell transformation, the synthesis and degradation of extracellular matrix, and the release of fibrotic mediators. Therefore, this article reviews the biological characteristics of miR-15a/16 and the molecular mechanisms and functions of their dysregulation in fibrotic diseases.

## 1. Introduction

MicroRNAs (miRNAs) are a class of endogenous, non-coding RNAs with lengths of about 18 to 22 nucleotides. miRNAs can directly degrade a target mRNA or inhibit its protein translation by completely or incompletely binding to the 3′ untranslated region (3′UTR), the coding region, or the 5′UTR of the target mRNA in order to play a negative role in regulating gene expression [1]. It is predicted that miRNAs can target more than 50 percent of human protein-coding genes [2] and affect hundreds of gene transcripts, thus participating in key processes, such as cell proliferation, transformation, invasion, migration, and apoptosis [3].

Except for a few miRNAs that have their own regulatory elements and can be transcribed independently of host genes [4,5], most of the miRNAs located in the genome are co-transcribed and expressed with the host gene. The classical pathway of their generation is that the genes encoding miRNAs in the nucleus are transcribed into primary miRNAs (pri-miRNAs) under the action of RNA polymerase II. pri-miRNA is re-cleaved into precursor miRNA (pre-miRNA) by the RNase Ⅲ/Drosha enzyme and its cofactor DGCR8. The pre-miRNA is released into the cytoplasm by exportin 5 and then processed into double-stranded RNA by the Dicer enzyme, HIV-1 TAR ribonucleic acid binding protein (TRBP), and Argonaute 2 (AGO2). One strand is degraded, while the other mature miRNA strand is loaded into the miRNA-induced silencing complex (RISC), which can bind incompletely to the miRNA-recognition elements (MRE) in the 3′UTR of the target gene, leading to post-transcriptional gene silencing or complete complementary binding leading to mRNA degradation [4,6].

Fibrosis refers to the pathological process by which the parenchymal cells of the affected organ become necrotic and the extracellular matrix (ECM) components in the tissue increase abnormally and decrease via degradation due to the dysfunction of various pathways, such as tissue repair and inflammation. If it continues to progress, the structure of the organ may be damaged and the performance of the organ decrease, even leading to organ failure [7]. As an important tumor suppressor gene cluster, miR-15a and miR-16 are known to regulate the tumor cell cycle, tumor proliferation, and tumor invasion. However, in recent years, studies have found that miR-15a/16 plays a strong regulatory role in fibroblast differentiation, extracellular matrix synthesis and degradation, and the release of fibrotic mediators and that it plays an important role in various fibrotic diseases. Therefore, this article reviews the biological characteristics of miR-15a/16 and its expression and regulatory mechanism in fibrotic diseases.

## 2. Biological Characteristics of the miR-15 Family

The miR-15 family is a class of miRNAs with a seed region of AGCAGC (AGC×2) starting from the second nucleotide of the 5′ end and the presence of common target genes, including miR-15a/b, miR-16-1, miR-16-2, miR-195, miR-424, and miR-497 [8,9]. In addition, miR-503 differs from the seed region of the typical miR-15 family only at the eighth nucleotide. miR-103, miR-107, and miR-646, whose first nucleotide sequence at the 5′ end is AGC×2, have all been classified as “extended” miR-15 family members (Table 1) [2,8].

Among the miR-15 family members, mature miR-16-1 has the same 5p sequence as miR-16-2 and is therefore indiscriminately referred to as miR-16-5p. miR-15a-5p and miR-15b-5p have only four nucleotide differences, but their seed region sequence is completely consistent, so they may share a large number of target genes. However, the seed sequences of miR-15a-3p and miR-15b-3p are very different, which may lead to functional differences between these two miRNA clusters [10]. miR-15a and miR-16 constitute a highly conserved miR-15a/16-1 gene cluster, which is located in the intron of the long non-coding RNA (lncRNA) Dleu2 gene on human chromosome 13 (13q14) and mouse chromosome 14 (14qC3) and is regulated by its promoter [11].

Various factors are involved in regulating the expression of miR-15a/16, such as genomic abnormalities, transcriptional regulation, epigenetic modifications, and lncRNA. Many studies have shown that histone acetylation modification in the promoter region of the Dleu2 gene can cause the dysregulation of miR-15a/16-1. Chen et al. [12] found that knockdown of histone deacetylase 3 (HDAC3) and its inhibitor could upregulate the acetylation of the miR-15a/16-1 promoter region and then increase its expression. Similarly, the transcription factors c-Myc and BSAP (B cell-specific activator protein) have been shown to directly bind to the Dleu2 gene and recruit HDAC3 to negatively regulate the Dleu2 promoter to inhibit the expression of miR-15a/16 [13,14]. In addition, Liu et al. [15] found that lncRNA SNHG12 was an upstream factor causing the dysregulation of miR-16 expression in colorectal cancer cells, and lncRNA PFAR was also proven to negatively regulate the expression of miR-15a, thus playing a role in promoting pulmonary fibrosis [16].

## 3. The Role and Mechanism of miR-15a/16 in Fibrogenesis

The multi-layer regulatory network of the interaction of miRNA key targets, downstream proteins, and transcription factors mediates miRNA’s leading role in the overall situation in the cell. It is based on the unique mechanism whereby miR-15a and miR-16 regulate the key transcriptome features in the process of tissue fibrosis by targeting several groups of mRNAs in the signaling pathway. These features include cell transformation, the synthesis and degradation of extracellular matrix, and the release of fibrotic mediators (Figure 1).

### 3.1. Regulation of Cell Transformation

Myofibroblasts, as the most important cell type for the synthesis of the components of the extracellular matrix, such as collagen, elastin, and fibronectin, are the key cells in various fibrotic diseases. The sources of myofibroblasts include [17]: (1) myofibroblasts transformed from resting fibroblasts by cytokines; (2) epithelial cells transformed from epithelial mesenchyme; and (3) CD34+ progenitor cells derived from bone marrow, which migrate into fibrotic tissues and differentiate phenotypically. It has been found that the phenotypic transformation of fibroblasts into myofibroblasts is inseparable from the action of cytokines.

Transforming growth factor-beta (TGF-β) is the most important cytokine in the process of fibrosis. After binding to its receptor, TGF-β can activate the downstream signal dependent on or independently of the smad protein and exerts a powerful pro-fibrotic function [18]. Zhu et al. [19] found that the expression of miR-16 was upregulated in hepatitis C virus-induced liver fibrosis and that highly expressed miR-16 could inhibit the expression of hepatocyte growth factor (HGF) and Smad7, thereby promoting liver fibrosis. Smad7 is an inhibitory protein of the TGF-β/Smad signaling pathway that can effectively inhibit the phosphorylation of signal activators (Smad2 and Smad3) and subsequent downstream signal activation [20]. However, other studies have shown that miR-16 can significantly inhibit the expression of smad2 [21], smad3 [22], and smad5 [23] in fibroblasts, as well as the phenotypic remodeling process of fibroblasts to myofibroblasts induced by TGF-β in many ways, thus playing an anti-fibrotic role.

### 3.2. Regulation of the Synthesis and Degradation of ECM

Fibrosis is the result of an imbalance between the synthesis and degradation of extracellular matrix components. Pan et al. [21] found that under the synergistic effects of TGF-β, phosphatidylinositol 3 kinase/protein kinase B (Akt), mitogen-activated protein kinase (MAPK)/extracellular signal-regulated kinase (ERK), and other signaling pathways, the activity of matrix metalloproteinases (MMPs), which degrade ECM components in myofibroblasts, was inhibited, and that the expression of the tissue inhibitor of metalloproteinase (TIMP) was increased. This results in the excessive deposition and reduced degradation of ECM components, such as collagen and proteoglycans. miR-16 can negatively regulate TGF-β signaling by targeting smad2, upregulating the expression of fibrinolysis-dependent matrix metallopeptidase (MMP-2, etc.), and inhibiting the secretion of ECM components, such as collagen type I and collagen type III [24,25].

In addition, miR-15a and miR-16 can also affect other signaling pathways related to ECM formation. Yao et al. [26] found that miR-16-5p inhibited the activation of the Notch signaling pathway by targeting Notch2, thereby increasing the expression of MMP-1 and MMP-8 in human skin fibroblasts and promoting the degradation of collagen and other ECM components. Chen et al. [27] found that the overexpression of miR-15a inhibited Yes-associated protein (YAP1), a key effector molecule in the Hippo pathway, and affected the activation of the downstream signal YAP1/TEAD/Twist, thereby inhibiting the synthesis of collagen and other ECM components in lung fibroblasts. miRNA-16 mimic administration can inhibit the expression of the rapamycin-insensitive component of rapamycin targeting protein (Rictor) in human lung fibroblasts, affecting the mammalian target of the rapamycin complex 2 (mTOR C2) signaling pathway, and downregulate the expression of ECM-related proteins, such as secreted protein acidic and rich in cysteine (SPARC), ultimately improving pulmonary fibrosis in vitro and in vivo [28].

### 3.3. Regulation of the Release of Fibrotic Mediators

Persistent chronic inflammatory responses caused by various stimuli often spontaneously evolve into fibrosis, and pro-fibrotic mediators are the main driving force in the progression from injury to fibrosis. Immune cells such as macrophages and different lymphocyte subsets at the site of injury can release a large number of pro-inflammatory cytokines and pro-fibrotic mediators, such as interleukin (IL)-1, IL-6, IL-8, tumor necrosis factor (TNF)-α, platelet-derived factor (PDGF), fibroblast growth factor (FGF), keratinocyte growth factor (KGF-1), etc., causing further organ damage and cell death and promoting the development of injury to fibrosis [29,30].

On the one hand, miR-16 can directly inhibit the release of inflammatory cytokines and other pro-fibrotic mediators. Liang et al. [31,32] found that miR-16 could activate the MAPK pathway (including p38 and ERK1/2) and inactivate the JNK pathway by targeting programmed cell death 4 (PDCD4) and then inhibit the expression of pro-inflammatory cytokines (IL-6, TNF-α, monocyte chemoattractant protein-1, and IL-1β) and promote the secretion of anti-inflammatory factor IL-10, thereby inhibiting inflammation and the formation of fibrous scar tissue in organs. In addition, miR-16 can regulate the interactions between macrophages and activated T cells by downregulating the expression of programmed cell death-ligand 1 (PD-L1), a key immunosuppressor controlling the activation of T cells. miR-16 can convert the M2 polarization of macrophages into M1 polarization, activate CD4+T cells, and reduce the secretion of inflammatory cytokines [33].

On the other hand, miR-16 can regulate the expression of fibrosis mediators, such as IL-1β and IL-18, by affecting the NOD-like receptor protein 3 (NLRsP3) inflammasome pathway. Yang et al. [34] found that in lung tissues and cells stimulated by lipopolysaccharide (LPS), highly expressed miR-16 could directly target Toll-like receptor 4 (TLR4), affect TLR4/ Nuclear factor Kappa B (NF-kB) activation, and inhibit the NLRP3 inflammasome pathway. It plays a protective role in LPS-induced lung tissue injury and fibrosis. Interestingly, miR-15a/b was found to be highly expressed in cardiomyocytes with myocardial ischemia/reperfusion injury and hypoxia/reoxygenation injury [35]. In addition, miR-15b can promote hypoxia/reoxygenation-induced cardiomyocyte pyroptosis and cytokine release by targeting Sirtuin3 (SIRT3) and activating the NLRP3 inflammasome, both of which aggravate inflammation and fibrosis response [36].

## 4. miR-15a/16 and Fibrotic Disease

miR-15a and miR-16 are abnormally upregulated or downregulated in fibrotic diseases and promote or inhibit various fibrotic diseases by targeting different genes (Table 2).

### 4.1. Liver Fibrosis

Liver fibrosis is the intersection of liver structural deterioration, which often eventually leads to the occurrence of liver cirrhosis and hepatocellular carcinoma. Its main pathological features are the activation and phenotypic transformation of hepatic stellate cells. Guo et al. [37] confirmed via microarray analysis that miR-15b/16 expression was significantly downregulated in activated hepatic stellate cells. Similarly, Liu et al. [38] found that the expression of miR-16 in patients with liver disease was significantly lower than that in healthy controls and that it was closely related to indicators of liver fibrosis. It was found that miR-16 directly combined with smad2/3/5 and lysine oxidase-like 1 (LOXL1) and that other target genes synergistically acted on the TGF-β pathway, a signaling pathway essential for hepatic stellate cell activation, which eliminates the phenotypic characteristics of myofibroblasts, including the inhibition of collagen and lipogenesis, and mediates the repair of liver fibrosis [21,40]. In addition, miR-16 can inhibit the apoptotic resistance of hepatic stellate cells by targeting Bcl-2, a key member of the mitochondrial apoptosis pathway, and its downstream protease linked signaling (caspase 1, 3, 8, and 9) [37,38,39]. Kim et al. [41] found that the downregulation of miR-16 caused the overexpression of guanine nucleotide-binding α-subunit12 (Gα12) protein in liver fibrosis. Gα12 promotes autophagy by coupling with an autophagy-related gene (ATG12-5) and activates hepatic stellate cells.

### 4.2. Pulmonary Fibrosis

Idiopathic pulmonary fibrosis (IPF) is a progressive, interstitial fibrotic lung disease characterized by cell proliferation, interstitial inflammation, and fibrosis. miR-16 has been shown to be a key regulatory factor in the progression of idiopathic pulmonary fibrosis and can be used as an indicator for the early diagnosis and treatment of IPF. Lacedonia et al. [42] found that lower levels of anti-fibrotic miRNAs (e.g., miR-16 and let-7d) and higher levels of pro-fibrotic miRNAs (e.g., miR-21) were detected in the serum exosomes of patients with IPF. Similarly, Xie et al. [43] found that miR-16 was downregulated in a mouse model of bleomycin-induced pulmonary fibrosis and targeted the regulation of cell apoptosis and inflammation-related pathways, such as Wnt signaling and TLR signaling, in the early stage of lung injury. miR-16 can target fibrosis-related pathways, such as the TGF-β signaling pathway, in the late stages of lung injury [43]. Other studies have also confirmed that miR-15a/16 can regulate the activation and apoptosis of fibroblasts, inflammation, and fibrosis in pulmonary fibrotic tissues by targeting Wnt ligands (including Wnt10b and Wnt3a [44]), TLR4 [34], autophagy-related genes 5 and 7 (ATG5/ATG7) [45], the Rictor/mTORC2 pathways [28], the lncRNA PFAR/YAP1/Twist axis [16,27], and other genes. In addition, Dehmel et al. [46] performed an miRNA analysis via flow cytometry sorting and purification of alveolar epithelial cell type II (ATII) and found that miR-16 was expressed abundantly in normal alveolar cells but that the expression of miR-16 was inhibited after TGF-β stimulation, while the expression of potential targets, such as MAP2K1, MAP2K4, JUN, and BCL2, was upregulated. Therefore, miR-16 can regulate cell activation pathways to maintain alveolar cell homeostasis and prevent the occurrence and development of lung injury, such as fibrotic lung diseases.

### 4.3. Cardiac Fibrosis

Myocardial fibrosis plays a key role in the occurrence and development of atrial structural remodeling and is a common pathological feature of cardiovascular diseases, such as hypertensive heart disease, myocardial infarction, heart failure, etc. Fang et al. [47] found that relatively stable circulating miR-15a was upregulated in diffuse myocardial fibrosis and affected cardiac-disease-related pathways, such as apoptosis, MAPK signaling, TGF-β, and other cytokine signaling pathways, meaning that it could be used as a biomarker and therapeutic target for the diagnosis of myocardial fibrosis. Porrello et al. [48] demonstrated that inhibiting the miR-15 family in the neonatal period can increase cardiomyocyte proliferation and improve left ventricular systolic function after myocardial infarction in adult hearts. Similarly, He et al. [49] found that miR-15a-5p could play a pro-fibrotic role by targeting Smad7 to regulate the expression of TGF-β/collagen type I in patients with heart failure and in LPS-stimulated myocardial fibrosis cell models. In hypertensive heart disease, miR-16, in combination with miR-19b, can downregulate the adrenergic receptor A 1a gene (ADRA1A) and significantly promote cardiac fibrosis and myocardial cell apoptosis in tissues [50]. However, miR-15a/b expression is downregulated and TGF-β and connective tissue growth factor (CTGF) expression is upregulated in diabetic hearts characterized by progressive myocardial fibrosis, mouse models, and high-glucose-induced cardiomyocytes, which can be reversed by restoring miR-15a/b expression, as it reduces the differentiation of cardiac fibroblasts and improves myocardial fibrosis [51].

### 4.4. Renal Fibrosis

Renal fibrosis, which mainly involves the glomerulus or tubulointerstitium, is a common complication of a variety of chronic kidney diseases. miR-16 is highly expressed in cells from all regions of normal kidneys [57]. Mohan et al. [52] found that the expression of miR-16 and miR-451 was upregulated in urinary exosomes in a diabetic rat model but downregulated in kidney tissues of this model. This may be due to the release of intracellular miR-16 into body fluids after renal tissue injury and rupture, which leads to the upregulation of the expression of miR-16 in urinary exosomes, supporting its protective role in renal tissues. In addition, the upregulation of miR-16 has been shown to inhibit inflammation-induced renal fibrosis and glomerular damage and to inhibit the activation of the TLR4 signaling pathway by targeting differential expression 2 (DEC2), thereby inhibiting renal tissue hyperplasia and mesangial cell proliferation, and alleviating the pathological symptoms of renal fibrosis [53]. Similarly, Sun et al. [54] found that low-dose paclitaxel could significantly improve renal function and reduce renal injury and tubulointerstitial fibrosis after subtotal renal resection by upregulating the expression of miR-15a, while miR-15a at least promoted the protective effect of paclitaxel on renal disease, to a certain extent, and was closely related to the TGF-β/Smad signaling pathway.

### 4.5. Other Fibrotic Diseases

Systemic sclerosis is an autoimmune fibrotic disease characterized by local or diffuse skin thickening and fibrosis of internal organs. Dolcino et al. [55] found that miR-16 could promote the apoptosis of endothelial cells, resulting in the release of intracellular miR-16 into the serum of patients with systemic sclerosis, and its expression was detected to be upregulated. Manesh et al. [56] found that miR-16 may affect the important pathogenic events of systemic sclerosis by targeting the nuclear proteins survivin and p53 that control cell cycle and apoptosis, inhibiting endothelial cell proliferation and promoting apoptosis. Similarly, it has been found that miR-16-5p can directly bind Smad3 [22] and Notch2 [26] to inhibit the activation and phenotypic transformation of fibroblasts and reduce the synthesis of collagen and other ECM and α-Smooth muscle actin(α-SMA) expressions, indicating that it has a direct anti-fibrotic effect.

## 5. Conclusions

Regulating the expression of miRNAs in target cells or tissues, that is, restoring the activity of anti-fibrotic miRNAs or inhibiting the function of pro-fibrotic miRNAs, may be an effective strategy to treat fibrotic diseases. For example, miR-29, as an important regulator of multiple fibrotic diseases, has been studied to evaluate the efficacy of an miR-29 mimic (Remlarsen) in phase 2 clinical trials. It has been found that the use of single or multiple doses of Remlarsen in normal healthy volunteers can limit the fibrogenesis of skin incisions and the formation of fibrous scar tissue in pulmonary fibrosis [58]. In a mouse model of obstructive nephropathy, Qin et al. [59] used ultrasound microbubble-mediated gene transfer technology to deliver plasmids overexpressing miR-29 to the lesion kidney well, thereby effectively preventing the progression of renal fibrosis. Meanwhile, miR-15a/16, which regulates fibrosis, could also reduce bone metastasis and achieve anticancer efficacy when combined with a vector and injected into a mouse model of a prostate cancer xenograft via the tail vein [60,61].

Although clinical experiments and animal models have shown the bright prospects of miRNA in the treatment of fibrotic diseases, the clinical research on the transformation of miRNA to produce therapeutic drugs is still in its infancy, and there are still many obstacles to their clinical use. Targeting the key problems that miRNAs are unstable and easily degraded by serum and need to achieve tissue-specific delivery, various viral and non-viral delivery systems, such as non-collagen proteins and liposomes, have been developed to improve the stability of their delivery [62,63]. Moreover, the targeted delivery of specific fibrotic types can be achieved by binding targeted conjugates, such as glycoconjugates, polypeptides, and antibodies [64]. However, the safety and reliability of these methods still need to be further studied.

In summary, miR-15a/16 is an important molecule in regulating the occurrence and development of fibrotic diseases. miR-15a/16 can inhibit the proliferation and phenotypic transformation of fibroblasts, inhibit ECM synthesis, promote ECM degradation, induce cell apoptosis, and affect the release of fibrotic mediators by targeting the fibrotic signaling pathway, the inflammatory signaling pathway, the cell proliferation and apoptosis signaling pathway, collagen synthesis, and other related genes, thereby weakening liver fibrosis, pulmonary fibrosis, and renal fibrosis. It can also accelerate the progression of myocardial fibrosis by activating the signal transduction of the pro-fibrotic factor TGF-β and by interfering with normal physiological processes, such as cell growth and apoptosis. As there are differences in the target genes between different cell types and in the pathogeneses and immune cell responses between different fibrotic diseases, the dysregulation, mechanism, and function of miR-15a/16 in the process of fibrosis are different in different tissues and organs. Therefore, further exploring the target genes of miR-15a/16 in different cell types and the relationships between miR-15a/16 and fibrotic diseases will help to clarify the pathogeneses of fibrotic diseases so as to provide new ideas for their treatment.

## Figures and Tables

**Figure 1 ijms-23-16041-f001:**
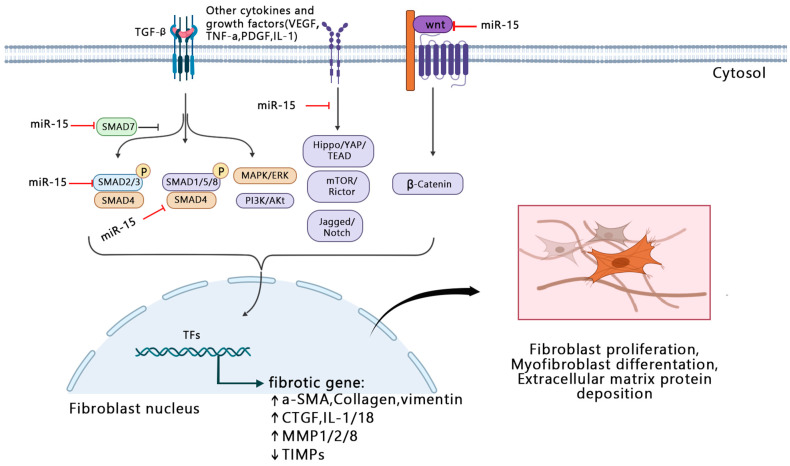
Possible regulatory network model of miR-15a/16 in the progression of fibrosis. ↑ indicates upregulation; ↓ indicates downregulation. TGF-β: Transforming growth factor-beta; VEGF: Vascular endothelial growth factor; TNF-a: Tumor necrosis factor-a; PDGF: Platelet derived growth factor; IL: Interleukin; MAPK: Mitogen-activated protein kinase; ERK: Extracellular signal-regulated kinase; PI3K: Phosphatidylinositol 3 kinase; YAP: Yes-associated protein; TEAD: Transcriptional enhancer associate domain; mTOR: mammalian Target of the rapamycin; Rictor: Rapamycin targeting protein; TFs: Tissue factors; a-SMA: α-Smooth muscle actin; CTGF: Connective tissue growth factor; MMP: Matrix metalloproteinase; TIMPs: Tissue inhibitor of metalloproteinases. Created with BioRender.com.

**Table 1 ijms-23-16041-t001:** Gene sequences of miR-15 family members.

miRNA	The First Nucleotide of a Mature miRNA Sequence	Seed Regions	The Last Nucleotide of the Seed Sequence	The Remaining Sequence of the Mature miRNA
hsa-miR-15a-5p	U	AGCAGCA	C	AUAAUGGUUUGUG
hsa-miR-15b-5p	U	AGCAGCA	C	AUCAUGGUUUACA
hsa-miR-15a-3p	C	CAGGCCA	U	AUUGUGCUGCCUCA
hsa-miR-15b-3p	C	GAAUCAU	U	AUUUGCUGCUCUA
hsa-miR-16	U	AGCAGCA	C	GUAAAUAUUGGCG
hsa-miR-195	U	AGCAGCA	C	AGAAAUAUUGGC
hsa-miR-424	C	AGCAGCAAU	U	CAUGUUUUGAA
hsa-miR-497	C	AGCAGCA	C	ACUGUGGUUUGU
hsa-miR-503	U	AGCAGCG	G	GAACAGUUCUGCAG

**Table 2 ijms-23-16041-t002:** Target genes, functions, and relationship with disease of miR-15a/16.

miRNA	Disease Type	Expression	Target Gene	Biological Processes Affected	Study Model/Cell	Reference
miR-16	Liver fibrosis	Upregulation	HGF and Smad7	Affects the TGF-β/Smad signaling pathway and promotes liver fibrosis	QSG-7701 cells and Huh-7.5.1 cells	[19]
miR-16	Liver fibrosis	Downregulation	Smad2/3	Inhibits the TGF-β/Smad pathway	HSCs	[21]
miR-16	Skin fibrosis	Downregulation	Smad3	Inhibits the TGF-β/Smad pathway	Exosomes from HaCaT cells and HDFs	[22]
miR-16	Systemic sclerosis	Downregulation	Notch2	Inhibits Notch signaling and reduces the synthesis of collagen and other ECM components	HDFs and HSFs	[26]
miR-15a	Pulmonary fibrosis	Downregulation	YAP1/Twist	Modulates the lncRNA PFAR/YAP1/Twist axis to inhibit cell activation and ECM deposition	Primary lung fibroblasts	[16,27]
miR-16	Pulmonary fibrosis	Downregulation	Rictor	Inhibits the mTORC2 signaling pathway, downregulates SPARC expression, and reduces ECM synthesis	Human fetal lung fibroblast (HFL-1) cells	[28]
miR-16	Pulmonary fibrosis	Downregulation	TLR4	Inhibits the TLR4/NF-kB pathway and the NLRP3 inflammasome signaling pathway	A549 cells and 293 T cells	[34]
miR-16	Liver fibrosis	Downregulation	Bcl-2	Affects the mitochondrial apoptosis pathway and induces apoptosis	HSCs	[37,38,39]
miR-16	Liver fibrosis	Downregulation	LOXL1	Inhibits Smad2/3 phosphorylation and inhibits the TGF-β/Smad pathway	LX2 cells	[40]
miR-16	Liver fibrosis	Downregulation	Gα12	Affects its binding to ATG12-5 and induces autophagy	HSCs	[41]
miR-16	Pulmonary fibrosis	Downregulation	ND	ND	Serum exosomes of IPF patients	[42]
miR-16	Pulmonary fibrosis	Downregulation	Bcl2, NF-kB, IL-1, JNK, Smad3/7, COL-I, and TGFβ-R	Affects apoptosis, Wnt signaling, TLR signaling, and the TGF-β/Smad pathway	Bleomycin mouse model of IPF	[43]
miR-16	Pulmonary fibrosis	Downregulation	Wnt ligands (including Wnt10b and Wnt3a)	Attenuates the Wnt signaling pathway, inhibits fibroblast transformation, and induces epithelial senescence	HBEC EVs and mouse models of BLM-induced lung fibrosis	[44]
miR-16	Pulmonary fibrosis	Downregulation	ATG5/7	Affects mitochondrial dysfunction and cellar senescence	MRC-5 cells and endothelial cells	[45]
miR-16	Pulmonary fibrosis	Downregulation	MAP2K1, MAP2K4, JUN, and BCL2	Inhibits the TGF-β/Smad pathway	A549 cells and primary murine ATII cells	[46]
miR-15a	Cardiac fibrosis	Upregulation	ND	Promotes the MAPK and TGF-β/Smad pathways	Serum of diffuse myocardial fibrosis patients	[47]
miR-15a, miR-16	Cardiac fibrosis	Upregulation	ND	Represses cardiomyocyte proliferation	C57BL/6 mice	[48]
miR-15a	Cardiac fibrosis	Upregulation	SMAD7	Promotes the TGF-β/Smad pathway	Lipopolysaccharide (LPS)-stimulated H9C2 cells	[49]
miR-16	Hypertensive heart disease	Upregulation	ADRA1A	Promotes cardiomyocyte apoptosis	Primary myocardial cells and H9C2 cells	[50]
miR-15a	Cardiac fibrosis	Downregulation	ND	Affects the TGF-β/Smad and p53 pathways	Human primary cardiac fibroblasts and Type 2 diabetic mice	[51]
miR-16	Renal fibrosis	Downregulation	MMP-9 and IL-6	Affects the fibrotic/inflammatory genes in the kidney tissue	Urinary exosomes (UEs) of diabetic rats	[52]
miR-16	Renal fibrosis	Downregulation	DEC2	Inhibits the TLR4 signaling pathway	SV40 MES-13 cells and Fcgamma receptor II-b-deficient (Fcgr2b−/−) mouse	[53]
miR-15a	Renal fibrosis	Downregulation	ND	Affects the TGF-β/Smad signaling pathway	Remnant kidney model and NRK-52E cells	[54]
miR-16	Systemic sclerosis	Upregulation	ND	Induces apoptosis	Serum of systemic sclerosis patients	[55]
miR-16	Systemic sclerosis	Downregulation	Survivin and p53	Inhibits cell proliferation and promotes cell apoptosis	HSFs	[56]

Abbreviations—HGF: hematopoietic growth factor; LOXL1: lysyl oxidase-like 1; YAP: yes-associated protein; Rictor: rapamycin-insensitive companion of mTOR; SPARC: secreted protein acidic and rich in cysteine; ADRA1A: human adrenergic alpha-1A receptor; HBEC: human bronchial epithelial cell; EVs: extracellular vesicles; HDFs: human dermal fibroblasts; HSCs: hepatic satellite cells; HSFs: human skin fibroblasts; ND: not determined.

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
