# Peer review of "The Molecular Mechanisms and Function of miR-15a/16 Dysregulation in Fibrotic Diseases"

_ijms, 2022, doi:10.3390/ijms232416041_

Round 1

Reviewer 1 Report

Many novel miRNAs are now associated with fibrosis, both organ-specific and systemic, as in the prototypical fibrotic disease systemic sclerosis. Dysregulated miRNAs are amenable to therapeutic modulation. In this paper, Da-Da Wen et al reviews the abnormal expression level and the molecular mechanism of miR-15a/16 in liver, lung, heart, kidney, and other fibrotic disease. Therefore, they suggested that miR-15a/16 could be developed as a biomarker for the diagnosis of fibrosis and also as a therapeutic target for anti-fibrosis.

This should be a brief, very relevant and interesting review article. It would be nice if the authors could give a small summary of the role of miRNAs in fibrosis (pro-fibrotic or anti-fibrotic) and the possibility of these being used as therapeutic targets and highlight the advantages of miR-15a/16. And it is recommended that authors have English speakers or professionals revise their manuscripts to make grammar and sentence trends easier for readers to navigate and understand.

Author Response

Response to Reviewer 1 Comments
Point 1: It would be nice if the authors could give a small summary of the role of miRNAs in fibrosis (pro-fibrotic or anti-fibrotic) and the possibility of these being used as therapeutic targets and highlight the advantages of miR-15a/16.

Response 1: We sincerely appreciate your careful review and these constructive suggestions!

Based on your kind suggestions, we have rewritten the Conclusions section as follows:

“In summary, miR-15a/16 is an important molecule in regulating the occurrence and development of fibrotic diseases. miR-15a/16 can inhibit the proliferation and phenotypic transformation of fibroblasts, inhibit ECM synthesis, promote ECM degradation, induce cell apoptosis, and affect the release of fibrotic mediators by targeting the fibrosis signal pathway, inflammation signal pathway, cell proliferation and apoptosis signal pathway, collagen synthesis, and other related genes, thereby weakening liver fibrosis, pulmonary fibrosis, and renal fibrosis. It can also accelerate the progression of myocardial fibrosis by activating the signal transduction of the pro-fibrotic factor TGF-β and by interfering with normal physiological processes such as cell growth and apoptosis. Because there are differences in the target genes in different cell types and the pathogenesis or immune cell response between different fibrosis diseases, the dysregulation, mechanism, and function of miR-15a/16 in the process of fibrosis in different tissues and organs are different. Therefore, further exploring the target genes of miR-15a/16 in different cell types and the relationship between miR-15a/16 and fibrotic diseases will help to clarify the pathogenesis of fibrotic diseases so as to provide new ideas for the treatment of fibrotic diseases.” (Page 11, Line 297-312)

Point 2: it is recommended that authors have English speakers or professionals revise their manuscripts to make grammar and sentence trends easier for readers to navigate and understand.

Response 2: We sincerely thank you for these comments. Based on your kind suggestion, we asked a professional native speaker of English to revise my manuscript.

Reviewer 2 Report

This manuscript summarizes the molecular mechanisms and the role of miR-15 and miR-16 in fibrotic diseases. This review is interesting, although the information on each reference is scarce.  

·      In the summary of each reference mentioned in the manuscript, must detail more information about the experimental design and sample type. For example, when the manuscript describes references 42 and 44, it is important to indicate that the miRNAs are derived from exosomes and extracellular vesicles respectively. Therefore, please add a column in table 2 with the information on the sample type that was used in each study.

·      It is important to be precise in each reference if the study is about miR15a/16 as a cluster or as independent miRNAs. For example, line 120, 142,153, and 165 says miR15a/miR16 but the examples describe miRNA-15 or miRNA-16 as independent miRNAs.

·      Please check carefully that all the references mentioned in the manuscript are also described in table 2. For example, references 42 and 43 are not present.

·      Add a paragraph with the following recent reference

Dehmel S, Weiss KJ, El-Merhie N, Callegari J, Konrad B, Mutze K, Eickelberg

O, Königshoff M, Krauss-Etschmann S. microRNA Expression Profile of Purified Alveolar Epithelial Type II Cells. Genes (Basel). 2022 Aug 10;13(8):1420. DOI:10.3390/genes13081420. PMID: 36011331; PMCID: PMC9407429.

And as minor corrections:

·      Review all the text, because it has many mistakes in the spaces between words. Also, correct some typographic mistakes, for example, on page 4, line 127 “fibroblasts.rly?” or line 139,” b-FGF”?.

·      Regarding table 2. Remove the plural form in the subtitles (Disease type and Expression)

·       In the column of “biological processes affected” edit the information because it is overlapped

Author Response

Response to Reviewer 2 Comments

Point 1: In the summary of each reference mentioned in the manuscript, must detail more information about the experimental design and sample type. For example, when the manuscript describes references 42 and 44, it is important to indicate that the miRNAs are derived from exosomes and extracellular vesicles respectively. Therefore, please add a column in table 2 with the information on the sample type that was used in each study.

Response 1: We sincerely appreciate your careful review and these constructive suggestions! Based on your kind suggestion, we have added the information on the sample type that was used in each study in the table 2 as follows:

miRNA

Disease type

Expression

Target gene

Biological processes affected

Study model/cell

reference

miR-16

Liver fibrosis

Upregulation

HGF and

 Smad7

Affects the TGF-β/Smad signaling pathway and promotes liver fibrosis

QSG-7701 cells and Huh-7.5.1 cells

[19]

miR-16

Liver fibrosis

Downregulation

Smad2/3

Inhibits the TGF-β/Smad pathway

HSCs

[21]

miR-16

Liver fibrosis

Downregulation

Bcl-2

Affects the mitochondrial apoptosis pathway and induces apoptosis

HSCs

[37,38,40]

miR-16

Liver fibrosis

Downregulation

LOXL1

Inhibits Smad2/3 phosphorylation and inhibits the TGF-β/Smad pathway

LX2 cells

[39]

miR-16

Liver fibrosis

Downregulation

Gα12

Affects its binding to ATG12-5 and induces autophagy

HSCs

[41]

miR-15a

Pulmonary fibrosis

Downregulation

YAP1/Twist

Modulates the lncRNA PFAR/ YAP1/Twist axis to inhibit cell activation and ECM deposition

Primary lung fibroblasts

[16,27]

miR-16

Pulmonary fibrosis

Downregulation

Rictor

Inhibits the mTORC2 signaling pathway, downregulates SPARC expression and reduces ECM synthesis

Human fetal lung fibroblast (HFL-1) cells

[28]

miR-16

Pulmonary fibrosis

Downregulation

TLR4

Inhibits the TLR4/NF-kB pathway and the NLRP3 inflammasome signaling pathway

A549 cells and 293 T cells

[34]

miR-16

Pulmonary

fibrosis

Downregulation

ND

ND

Serum exosomes of IPF patients

[42]

miR-16

Pulmonary

fibrosis

Downregulation

Bcl2, NF-kB, IL-1, JNK, Smad3/7, COL-I, and TGFβ-R

Affects apoptosis, Wnt signaling, TLR signaling, and the TGF-β/Smad

pathway

Bleomycin mouse model of IPF

[43]

miR-16

Pulmonary

fibrosis

Downregulation

Wnt ligands(including Wnt10b and Wnt3a)

Attenuates the Wnt signaling pathway, inhibits fibroblast transformation, and induces epithelial senescence

HBEC EVs and mouse models of BLM-induced

lung fibrosis

[44]

miR -16

Pulmonary fibrosis

Downregulation

ATG5/7

Affects mitochondrial dysfunction and cellar senescence

MRC-5 cells and endothelial cells

[45]

miR -16

Pulmonary fibrosis

Downregulation

MAP2K1, MAP2K4, JUN and BCL2

Inhibits the TGF-β/Smad pathway

A549 cells and primary murine ATII cells

[46]

miR -15a

Cardiac fibrosis

Upregulation

ND

Promotes the MAPK and TGF-β/Smad pathways

Serum of diffuse myocardial fibrosis patients

[47]

miR - 15a, miR - 16

Cardiac fibrosis

Upregulation

ND

Represses cardiomyocyte

proliferation

C57BL/6 mice

[48]

miR - 15a

Cardiac fibrosis

Upregulation

SMAD7

Promotes the TGF-β/Smad pathway

Lipopolysaccharide (LPS) stimulated H9C2 cells

[49]

miR-16

Hypertensive heart disease

Upregulation

ADRA1A

Promotes cardiomyocyte apoptosis

Primary myocardial cells and H9C2 cells

[50]

miR - 15a

Cardiac fibrosis

Downregulation

ND

Affects the TGF-β/Smad and p53 pathways

Human primary cardiac fibroblasts and Type 2 diabetic

mice

[51]

miR-16

Renal fibrosis

Downregulation

MMP-9 and IL-6

Affects the fibrotic/inflammatory genes in the kidney tissue

Urinary exosomes (UEs) of diabetic rats

[53]

miR-16

Renal fibrosis

Downregulation

DEC2

Inhibits the TLR4 signaling pathway

SV40 MES-13 cells and Fcgamma receptor II-b-deficient (Fcgr2b−/−) mouse

[54]

miR - 15a

Renal fibrosis

Downregulation

ND

Affects the TGF-β/Smad signaling pathway

Remnant kidney model and NRK-52E cells

[55]

miR-16

Skin fibrosis

Downregulation

Smad3

Inhibits the TGF-β/Smad pathway

Exosomes from HaCaT cells and HDFs

[22]

miR-16

Systemic

sclerosis

Downregulation

Notch2

Inhibits Notch signaling and reduces the synthesis of collagen and other ECM components

HDFs and HSFs

[26]

miR-16

Systemic

sclerosis

Upregulation

ND

Induces apoptosis

Serum of systemic

sclerosis patients

[56]

miR-16

Systemic

sclerosis

Downregulation

Survivin and

p53

Inhibits cell proliferation and promotes cell apoptosis

HSFs

[57]

Point 2: It is important to be precise in each reference if the study is about miR15a/16 as a cluster or as independent miRNAs. For example, line 120, 142,153, and 165 says miR15a/miR16 but the examples describe miRNA-15 or miRNA-16 as independent miRNAs.

Response 2: We sincerely thank you for these comments. We have revised the sentence in the original Line 120, 142, 153, and 165 as follows:

“In addition, miR-15a and miR-16 can also affect other signaling pathways related to ECM formation.” (Page 4, Line 120)

“On the one hand, miR-16 can directly inhibit the release of inflammatory cytokines and other pro-fibrotic mediators.” (Page 4, Line 142)

“On the other hand, miR-16 can regulate the expression of fibrosis mediators such as IL-1β and IL-18 by affecting the NOD-like receptor protein 3 (NLRsP3) inflammasome pathway.” (Page 4, Line 153)

“miR-15a and miR-16 are abnormally upregulated or downregulated in fibrotic dis-eases and promote or inhibit various fibrotic diseases by targeting different genes.” (Page 5, Line 165)

Point 3: Please check carefully that all the references mentioned in the manuscript are also described in table 2. For example, references 42 and 43 are not present.

Response 3: We sincerely thank you for these comments. Based on your suggestions, we have supplemented and refined the references mentioned in this article in Table 2.

Point 4: Add a paragraph with the following recent reference.

Response 4: We sincerely thank you for this comment. We have cited your recommended references in the pulmonary fibrosis section as follows:

“In addition, Dehmel et al. [46] performed a miRNA analysis via flow cytometry sorting and purifying alveolar epithelial cell type II (ATII) and found that miR-16 was expressed abundantly in normal alveolar cells, but the expression of miR-16 was inhibited after TGF-β stimulation, and the expression of potential targets such as MAP2K1, MAP2K4, JUN, and BCL2 was upregulated. Therefore, miR-16 can regulate cell activation pathways to maintain alveolar cell homeostasis and prevent the occurrence and development of lung injury such as fibrotic lung diseases.” (Page 8, Line 218-225)

Point 5: And as minor corrections:

1、Review all the text, because it has many mistakes in the spaces between words. Also, correct some typographic mistakes, for example, on page 4, line 127 “fibroblasts.rly?” or line 139,” b-FGF”?.

2、Regarding table 2. Remove the plural form in the subtitles (Disease type and Expression).

3、In the column of “biological processes affected” edit the information because it is overlapped

Response 5: We sincerely appreciate your careful review and these constructive suggestions!  Based on your kind suggestions, we have corrected some typographic mistakes and removed the plural form in the subtitles (Disease type and Expression).

For the column of “biological processes affected” section, we have re-edited the information.
